# Tackling Prostate Cancer with Theranostic E5B9-Bombesin Target Modules (TMs): From Imaging to Treatment with UniCAR T-Cells

**DOI:** 10.3390/ijms26062686

**Published:** 2025-03-17

**Authors:** Liliana R. Loureiro, Susan Pike, Melinda Wuest, Cody N. Bergman, Kira R. JØrgensen, Ralf Bergmann, Anja Feldmann, Frank Wuest, Michael Bachmann

**Affiliations:** 1Institute of Radiopharmaceutical Cancer Research, Helmholtz-Zentrum Dresden-Rossendorf (HZDR), 01328 Dresden, Germany; l.loureiro@hzdr.de (L.R.L.); a.feldmann@hzdr.de (A.F.); 2Department of Oncology, University of Alberta, Cross Cancer Institute, Edmonton, AB T6G 1Z2, Canada; srichter@ualberta.ca (S.P.); mwuest@ualberta.ca (M.W.); cnbergma@ualberta.ca (C.N.B.); krjorgen@ualberta.ca (K.R.J.); 3Cancer Research Institute of Northern Alberta (CRINA), University of Alberta, Edmonton, AB T6G 2E1, Canada; 4Department of Biophysics and Radiation Biology, Semmelweis University, 1094 Budapest, Hungary; ralf.bergmann@med.semmelweis-univ.hu; 5German Cancer Consortium (DKTK), Partner Site Dresden, and German Cancer Research Center (DKFZ), 69120 Heidelberg, Germany; 6National Center for Tumor Diseases Dresden (NCT/UCC), German Cancer Research Center (DKFZ), 69120 Heidelberg, Germany; 7Faculty of Medicine and University Hospital Carl Gustav Carus, TU Dresden, 01307 Dresden, Germany

**Keywords:** UniCAR T-cell therapy, bombesin, theranostics, prostate cancer, PET

## Abstract

Target modules (TMs), intermediate molecules required for UniCAR T-cell therapy, are promising molecules for immunotheranostic approaches. In the current work, we developed TMs containing a monomeric or dimeric form of the antagonist bombesin peptide (BBN2) and assessed their potential for diagnostic imaging using positron emission tomography (PET) as well as immunotherapy in combination with UniCAR T-cells to target and image GRPR expression in prostate cancer. Synthesized monomeric and dimeric BBN2 TMs retained binding to GRPR in vitro. Both BBN2 TMs specifically activated and redirected UniCAR T-cells to eradicate PC3 and LNCaP cancer cells with high efficiency and in a comparable manner. UniCAR T-cells retained a non-exhausted memory phenotype favorable to their persistence and fitness. The ^68^Ga-labeled BBN2 TMs showed proof-of-target towards GRPR in PC3 and LNCaP xenografts with similar uptake profiles for both BBN2 TMs in dynamic PET experiments. Clearance occurred exclusively through renal elimination. A tremendously increased in vivo metabolic stability of the BBN2 TMs was observed compared to their counterparts without E5B9. Both monomeric and dimeric BBN2 TMs represent novel and promising immunotheranostic tools for application in prostate cancer with exceptionally high in vivo metabolic stability.

## 1. Introduction

Chimeric Antigen Receptor (CAR) T-cell therapy is a recognized groundbreaking type of immunotherapy that relies on the modification of T-cells from a patient to augment their ability to recognize, target, and eliminate cancer cells. This approach has demonstrated significant success, particularly in treating hematological malignancies [1]. Despite such achievements, applying this therapy to solid tumors has been challenging due to the complex and immunosuppressive tumor microenvironment. To optimize and expand the application of this innovative treatment, alternative adapter CAR therapies have been developed. Among them, the UniCAR platform developed by our group additionally brings controllability, flexibility, and safety features [2,3,4] (NCT04230265, NCT04633148). It relies on a bimodular adapter molecule called target module (TM), which is essential and responsible for the specific linkage of UniCAR T-cells to tumor cells [5,6,7]. A TM is therefore composed of the UniCAR peptide epitope (E5B9) linked to a binding moiety that specifically recognizes a tumor-associated antigen (TAA) on the target cells [8,9]. The UniCAR system has been proven to be broadly and successfully applicable and, in addition, the TMs have demonstrated their great potential as immunotheranostic tools [10,11]. So far, we have developed a series of TMs with different formats ultimately leading to different half-lives and binding properties [12,13]. Particularly effective examples of theranostic TMs are the anti-PSCA-IgG4-TM [10], the anti-FAP-IgG4-TM [11], the nanobody-based anti-EGFR-TM [14], and the PSMA PLT-TM [15].

The gastrin-releasing peptide receptor (GRPR) is a well-established target in preclinical and clinical cancer research [16,17,18]. Its overexpression in, e.g., prostate [19,20], breast [21], and lung cancers [22] has been well documented as playing a role in tumor cell growth, proliferation, angiogenesis, and migration [16]. Currently, GRPR is targeted with peptide ligand bombesin and its analogs, representing the molecular basis for targeted drug delivery [23]. In the last decades, numerous radiolabeled bombesin analogs for cancer diagnosis and therapy have been described, with ^68^Ga-RM2 [24,25], ^68^Ga-NeoBOMB1 [26,27], and ^64^Cu-SAR-BBN [28] presently being utilized in clinical trials. Our group has developed ^18^F-, ^68^Ga-, and ^44g^Sc-labeled BBN2 peptides [29,30,31], representing an intrinsically metabolically stable bombesin antagonist suitable for targeting prostate and breast cancer.

This study aims to develop new immunotheranostic TMs using BBN2 peptides for diagnostic imaging using positron emission tomography (PET) and UniCAR T-cell therapy to specifically target GRPR in prostate cancer. For that, TMs with the BBN2 peptide motif were synthesized as monomers and dimers, and evaluated in vitro for their immunotherapeutic potential and in vivo for their stability and imaging of GRPR-expressing prostate tumors in mice.

## 2. Results

### 2.1. Targeting GRP Receptor Using UniCAR T-Cells Redirected by BBN2 TMs

Given the upregulation and relevance of the GRPR in cancer, this work aims to develop an immunotheranostic approach using TMs derived from the bombesin antagonist BBN2 in monomeric and dimeric forms. We envision these TMs to be suitable for imaging, and potentially radiotherapy, as well as for immunotherapy in combination with UniCAR T-cells (Figure 1A). Synthesis of peptide-based BBN2 TMs was accomplished using a combination of automated and manual solid-phase peptide synthesis (SPPS). Figure 1B summarizes their chemical structures.

For the monomeric BBN2 TM, metabolically stabilized BBN2 was generated on Rinkamide MBHA resin, the *N*-terminus was coupled to Fmoc-Lys(N_3_)-OH, and the spacer Fmoc-Ava-OH was used to generate a clickable peptide. Macrocyclic chelator SCN-Bn-NOTA (denoted as NOTA) was reacted with the *N*-terminus of the peptide in solution for future radiolabeling with Ga-68. The E5B9 epitope was synthesized on resin in isolated yields of 61% and modified with the click moiety DBCO-PEG_4_-Mal in solution via its Cys^9^-residue to yield DBCO-PEG4-maleimide-E5B9 (57%). Copper-free strain-promoted azide–alkyne cycloaddition (SPAAC) gave access to E5B9-NOTA-BBN2 (monomeric BBN2 TM; 49.3% yield). The respective ‘click-connector’ is outlined in Figure 1B. The dimeric BBN2 TM also made use of the DBCO-modified DBCO-PEG4-maleimide-E5B9 but was then coupled with a NOTA-modified dimeric BBN2 in the same SPAAC reaction with a 62% isolated yield. To ensure that the E5B9 tag is available on both TMs for effective binding to the UniCAR T-cells via the 5B9 scFv expressed on their surface, an ELISA was carried out and the binding of the 5B9 mAb to the E5B9 was successfully corroborated (Appendix A).

### 2.2. Therapeutic Evaluation of BBN2 TMs in Combination with UniCAR T-Cells

To evaluate the functionality and therapeutic potential of the novel BBN2 TMs, we focused on PC3 and LNCaP prostate cancer cell lines. Both cell lines express GRPR, and when co-cultured with UniCAR T-cells in the absence of TMs, only minimal killing (7 to 18%) was observed. In the presence of both BBN2 TMs, UniCAR T-cells were able to specifically and significantly kill both prostate cancer cell lines (50–60%) (Figure 2A). Furthermore, by titrating the TMs, an estimation of the killing efficiency was determined by calculating the half-maximal effective concentration (EC_50_). The results show very similar EC_50_ values for both TMs (30 nM) targeting both cell lines (Figure 2B). These low nanomolar values highlight the very efficient killing by redirected UniCAR T-cells using both BBN2 TMs. In line with previous control studies, the novel bombesin targeting TMs do not redirect UniCAR T-cells to GRPR-negative cells.

To further assess the efficacy of UniCAR T-cells targeting GRPR using BBN2 TMs, the release of pro-inflammatory cytokines responsible for the stimulation of other immune mechanisms and cells involved in the anti-tumor response was determined. UniCAR T-cells were co-cultured with either PC3 or LNCaP cells, in the absence or presence of E5B9-NOTA-BBN2 or E5B9-NOTA-(BBN2)_2_ TM. The results show that the main pro-inflammatory cytokines, such as IFN-γ, TNF-α, IL-2, and GM-CSF, were secreted by UniCAR T-cells exclusively in the presence of the BBN2 TMs (Figure 2C). As expected, in negative control conditions, where only target cells or only UniCAR T-cells were present, no release of any of the abovementioned cytokines was detected. Overall, an increased release of cytokines was observed for the combination of UniCAR T-cells with the dimeric BBN2 TM in comparison to monomeric BBN2 TM.

### 2.3. UniCAR T-Cell Phenotyping Based on Their Activation, Exhaustion, and Memory State

Envisioning a better assessment of the phenotypic state of the UniCAR T-cells after co-culturing with target cells and BBN2 TMs for 48 h, the expression of certain activation, exhaustion, and memory markers was determined using flow cytometry. UniCAR T-cell activation and exhaustion were estimated based on the expression of CD69, along with PD-1, LAG-3, and TIM-3. The results demonstrate that UniCAR T-cells are specifically activated and upregulate CD69 in the presence of GRPR-expressing cancer cells (PC3 or LNCaP) and the BBN2 TMs (Figure 3A). Along with this, in such conditions, the UniCAR T-cells also upregulate the expression of inhibitory/exhaustion markers (PD-1, LAG-3, and TIM-3), expressing mainly only one or two markers. Only around 12% of UniCAR T-cells co-express the three inhibitory markers. All of these data support the evidence that UniCAR T-cells are activated in a controlled way and exert cytotoxicity in a target-specific and TM-dependent manner (Figure 3B). In comparison and as expected, the expression levels of such activation and exhaustion markers were reduced in the conditions where the TMs are absent or only UniCAR T-cells are present.

Moreover, the memory phenotyping of UniCAR T-cells was assessed based on the expression of CD45RO and CD62L, and the T-cell subsets were defined as follows: CD45RO^−^CD62L^−^ as terminal effector T-cells (T_TE_), CD45RO^+^CD62L^−^ as effector memory T-cells (T_EM_), CD45RO^+^CD62L^+^ as central memory T-cells (T_CM_), and CD45RO^−^CD62L^+^ as naïve and stem cell-like memory T-cells (T_N_/T_SCM_) [32]. Figure 3C shows that UniCAR T-cells were predominantly in a central memory state in the control conditions in which only PC3 or LNCaP cancer cells were present. On the other hand, and as expected, the UniCAR T-cells shift to a mixture of central memory and effector memory states in the presence of BBN2 TMs. This is in line with all previous data, which corroborate the evidence that UniCAR T-cells undergo activation, release of pro-inflammatory cytokines, and killing of GRPR-expressing cancer cells exclusively in the presence of target cells and the respective BBN2 TMs.

### 2.4. In Vitro Binding Assessment of BBN2 TMs to GRPR

BBN2 TMs were tested in a competition binding assay against ^125^I-Tyr^4^-bombesin in PC3 cells (Figure 4). Native gastrin-releasing peptide (GRP) exhibited an IC_50_ value of 0.15 ± 0.05 nM. The monomeric BBN2 TM demonstrated a lower potency with an IC_50_ of 190 ± 0.04 nM in comparison to the dimeric BBN2 with 0.70 ± 0.10 nM.

### 2.5. ^68^Ga-Radiolabeling of Monomeric and Dimeric BBN2 TMs

E5B9-NOTA-BBN2 and E5B9-NOTA-(BBN2)_2_ TMs were radiolabeled with positron emitter Ga-68 (t_½_ = 68 min) via radiometal complexation of their respective chelator NOTA. An amount of 25 μg of peptide precursor was incubated with ^68^Ga-GaCl_3_ in NaOAc buffer to give a reaction pH of 4.5. ^68^Ga-incorporation was >95% after 17–20 min for the monomeric TM and >90% after 23–25 min for the dimeric TM confirmed by radio-TLC. Purification with solid-phase extraction (SPE) and reformulation in 10% EtOH/saline delivered ^68^Ga-NOTA-peptide TMs in high radiochemical purity of 95–99% (radio-HPLC, Appendix A) suitable for subsequent in vitro and in vivo testing. Total radiosynthesis time was 58.5 ± 10 min for the ^68^Ga-labeled monomeric BBN2 TM and 50 ± 8 min for the ^68^Ga-labeled dimeric BBN2 TM, while radiochemical yields (decay-corrected) were 57 ± 11% and 46 ± 12%, respectively.

### 2.6. In Vivo PET Imaging of ^68^Ga-Labeled Monomeric and Dimeric BBN2 TMs

PET experiments were performed in PC3 and xenografts. Figure 5A shows representative images of both ^68^Ga-BBN2 TMs at 60 min p.i. in comparison to the control ^68^Ga-NOTA-BBN2 lacking the E5B9 epitope [27]. The SUV*_mean,60min_* were all in a similar range in PC3 tumors with the monomeric ^68^Ga-BBN2 TM showing 0.44 ± 0.01 (n = 6), the ^68^Ga-NOTA-BBN2 0.46 ± 0.05 (n = 6), and the dimeric ^68^Ga-(BBN2)_2_ TM 0.51 ± 0.10 (n = 4). Uptake into LNCaP tumors was slightly lower with the monomeric ^68^Ga-BBN2 TM having a SUV*_mean,60min_* of 0.44 ± 0.11 (n = 3), the dimer ^68^Ga-(BBN2)_2_ TM 0.38 ± 0.03 (n = 4), and the control ^68^Ga-NOTA-BBN2 0.38 ± 0.02 (n = 3). Time–activity curves (TACs) are shown in Figure 5B. The novel ^68^Ga-BBN2 TMs also showed a shift towards a more renal elimination (Appendix A) compared to a higher hepatobiliary for the control ^68^Ga-NOTA-BBN2.

Specific binding to GRPR was analyzed with in vivo blocking studies using pre-dosing with GRPR-binding Ava-BBN2 (Figure 6) reaching 39 ± 9% (n = 3; *p* < 0.05) for the monomeric ^68^Ga-BBN2 TM in PC3 tumors and 43 ± 3% (n = 3; *p* < 0.01) for the dimeric ^68^Ga-(BBN2)_2_ TM in LNCaP tumors.

### 2.7. In Vivo Metabolic Stability of BBN2 TMs

Figure 7 summarizes the results obtained for the ^68^Ga-labeled BBN2 TMs in vivo metabolic stability in venous blood plasma. Over the time course of this study, the monomeric and dimeric TMs remained remarkably stable with 93 ± 1% (n = 3) intact ^68^Ga-peptide remaining after 60 min p.i. for the dimeric TM and 78 ± 7% (n = 3) for the monomeric TM.

For comparison, the in vivo metabolic stability of ^68^Ga-NOTA-BBN2 was added [27]. ^68^Ga-NOTA-Ava-BBN2 was more stable than native bombesin. However, compared to the BBN2 TMs, ^68^Ga-NOTA-BBN2 decreased over time to 38 ± 8% (n = 3) intact at 60 min p.i.

We also synthesized a dimeric BBN2 (^68^Ga-NOTA-(BBN2)_2_) to prove the stabilizing effect provided by the peptide dimerization and/or possibly by the E5B9 peptide. While ^68^Ga-NOTA-(BBN2)_2_ is more stable than ^68^Ga-NOTA-BBN2 at 5, 15, and 30 min p.i., the difference at 60 min p.i. is negligible with 37 ± 1% (n = 3) intact BBN2 dimeric TM.

## 3. Discussion

This study introduces new theranostic tools that combine the UniCAR peptide epitope E5B9 with the metabolically stable GRPR-targeting peptide bombesin antagonist BBN2 in monomeric and dimeric forms for imaging and treatment of prostate cancers. These molecules, herein named BBN2 TMs, have been characterized both in vitro and in vivo using different prostate cancer mouse models, including among others binding affinity assessment, target specificity, and therapeutic potential in combination with UniCAR T-cells, clearance pattern, and metabolic stability.

In vitro binding studies confirmed that monomeric and dimeric BBN2 TMs did not lose binding affinity to the GRPR due to the modification with the E5B9 peptide epitope. However, using dimeric BBN2 TMs increased the binding affinity to the GRPR to almost similar potencies as the natural ligand GRP. This impressive subnanomolar affinity to GRPR has not been accomplished with any other previously described BBN2 derivative [30]. Furthermore, both BBN2 TMs have promoted specific eradication of both GRPR-expressing prostate cancer cell lines with high efficiency and in a comparable manner. Such killing results were supported by the specific activation of the UniCAR T-cells upon cross-linkage with the BBN2 TMs and subsequent specific release of pro-inflammatory cytokines in such conditions. Here, the dimeric BBN2 TM seems to promote a slightly increased activation pattern and release of pro-inflammatory cytokines in comparison to the monomeric form. This observation may be explained by the increased valency of the dimeric BBN2 TM, which may be relevant in scenarios requiring a particular activation and cytokine induction threshold to obtain efficient killing. However, in this particular work, this does not seem to be the case as an efficient and comparable killing of the used prostate cancer cells was obtained.

In CAR T-cell therapy, the persistence and fitness of CAR T-cells represent key factors to achieve complete responses and prevent relapses in patients. These features are typically observed in non-exhausted memory T-cells [33]. Therefore, the memory phenotype and exhaustion pattern of CAR T-cells should be assessed. In this work, we evaluated the expression of established memory and exhaustion markers on UniCAR T-cells before and after co-culturing with target cells and BBN2 TMs. Exhausted T-cells are typically characterized by a sustained co-expression of inhibitory markers such as PD-1, LAG-3, and TIM-3 [34,35]. The present data demonstrate that our manufacturing protocol leads to the production of non-exhausted central memory UniCAR T-cells, which is in line with the multiple studies supporting the observations that this type of T-cell persists longer and exhibits greater anti-tumor activity in patients [33,34,35]. Upon co-culture with cancer target cells and the respective BBN2 TMs, a shift was observed in the memory phenotype of the UniCAR T-cells, in which a mixture of central and effector memory T-cells was more prevalent. A slightly increased co-expression of exhaustion markers can also be observed under these conditions. These observations were expected and in accordance with a typical phenotypical response to the challenge with cancer cells, representing the activation and cytotoxic activity of the UniCAR T-cells. It is worth mentioning that, upon such a challenge, the UniCAR T-cells are not completely exhausted (high co-expression of all inhibitory markers mentioned above) nor transit into an irreversible terminal effector phenotype (T_TE_), implying that their persistency and fitness are retained.

Rapid manual ^68^Ga incorporation under standard conditions provided access to the PET probes. ^68^Ga-labeled monomeric and dimeric BBN2 TMs were both efficient to image PC3 and LNCaP tumors similar to the radiolabeled BBN2 peptide itself [30]. Overall, androgen-dependent and androgen-independent prostate cancers can be successfully imaged with the novel BBN2 TMs. This is especially of relevance for PSMA-negative prostate cancers with regards to PSMA silent metastasis [36] as targeting GRPR would represent important additional therapy options for these patients.

The elimination pathway was altered for both BBN2 TMs compared to ^68^Ga-NOTA-BBN2 alone, favoring renal elimination via kidneys. Both BBN2 TMs are significantly larger but still under the required 60 kDa cutoff for renal glomerular filtration [37]. Renal elimination can also be attributed to the E5B9 peptide adding charged amino acids (Glu, Asp, and Lys) under physiological pH, facilitating renal elimination. The fusion of BBN2 with E5B9 epitope, leading to a renal elimination pattern and increased stability, tackles a known issue of bombesin radioligands suffering from high hepatobiliary excretion and hence unfavorable pharmacokinetics [23]. Overall, fast accumulation at the tumor site, rapid elimination from the bloodstream, and fast clearance in vivo were observed as prerequisites for the immunotherapeutic theranostic approach [15].

High target-to-background ratios for the BBN2 TMs confirm the specificity of the theranostic and give high contrast as well as low impact on healthy tissue. Persistent high GRPR affinity and specificity were validated using in vivo blocking studies, with a significant reduction in tumor uptake of both ^68^Ga-labeled monomeric and dimeric TMs when pre-dosed with Ava-BBN2. As visible from the PET images, non-specific uptake in normal tissue was not detected, which is essential to exclude/diminish on-target/off-tumor toxicity, which could lead to major side effects [38].

Poor plasma stability of peptides has led to shortcomings in their application as drug candidates. Due to stabilization techniques combined with their favorable characteristics, such as size, high sensitivity, rapid clearance, fast elimination, and low immunogenicity, peptides have become increasingly prominent drugs and theranostics for clinical application [39,40,41]. Meanwhile, ^68^Ga-NOTA-BBN2 per se is already an intrinsically stable bombesin analog with 40% intact peptide after 60 min p.i. [30], but the novel BBN2 TMs were revealed to have much higher in vivo metabolic stability. Notably, the dimeric BBN2 TM was found to be 93% intact after 60 min in mice. In addition, we also synthesized a BBN2 dimer without E5B9 to prove the increased stability due to the dimerization of peptides described elsewhere [42,43,44]. While it verified somewhat enhanced stability compared to the ^68^Ga-NOTA-BBN2 itself, the E5B9 epitope seems to boost in vivo metabolic stability even further. We propose that E5B9 safeguards BBN2 from proteolysis by peptidases (e.g., neutral endopeptidase, aminopeptidase) [23,45], therefore having a shielding effect. Additionally, short protein motifs have been described that stabilize peptides and small proteins. These motifs were rich in lysine residues [46] or proline residues [47]. The E5B9 peptide epitope carries two proline and one lysine residue, which may be key elements for improved in vivo stability. While this concept needs to be validated in the future, we have seen this phenomenon before with other ligands.

## 4. Materials and Methods

### 4.1. Materials and Cell Lines

All chemicals were obtained from MilliporeSigma (Oakville, ON, Canada). Peptides were synthesized via a combination of manual and automated Fmoc/*t*Bu solid-phase peptide synthesis (SPPS) using the Syro I peptide synthesizer (MultiSynTech/Biotage, Charlotte, NC, USA). A 50 mCi (1850 MBq) iThemba Laboratories ^68^Ge/^68^Ga generator from IsoSolutions Inc. (Vancouver, BC, Canada) was used as a ^68^Ga source. Mass spectra were recorded on an Agilent Technologies 1260 HPLC with G6130B MSD (LCMS ESI). Human prostate cancer cell lines PC3 (American Type Tissue Culture Centre, Manassas, VA, USA) were cultivated in Dulbecco’s Modified Eagle Medium (DMEM) supplemented with 10% heat-inactivated fetal bovine serum (FBS) and 1% penicillin/streptomycin from Invitrogen (Life Technologies Inc., Toronto, ON, Canada) and LNCaP (American Type Tissue Culture Centre, Manassas, VA, USA) in RPMI-1640 supplemented with 10% heat-inactivated FBS and 1% penicillin/streptomycin from Invitrogen (Life Technologies Inc., Toronto, ON, Canada). Both cell lines were kept at 37 °C in a humidified 5% CO_2_ atmosphere and routinely tested for mycoplasma by PCR. All animal studies were carried out according to the guidelines of the Canadian Council on Animal Care (CCAC) and approved by the Cross Cancer Institute Animal-Care Committee. In vivo studies were conducted using control BALB/c and male PC3 and LNCaP tumor-bearing NSG nude mice (body weight: 20−24 g, Charles River Laboratories). For tumor xenografts, about 7.5 × 10^6^ PC3 in 200 μL PBS or 20 × 10^6^ LNCaP cells in 200 μL of PBS/Matrigel (50/50) were injected into the shoulder of male nude NSG mice subcutaneously. After 3−5 weeks, PC3 and LNCaP tumors reached ∼300−500 mm^3^ in size and were used for the experiments described. Additional information on other general materials can be found in the Appendix A.

### 4.2. UniCAR T-Cells—T-Cell Isolation and Genetic Modification

Buffy coats (German Red Cross, Dresden, Germany) were used to isolate peripheral blood mononuclear cells (PBMCs) by density gradient centrifugation with Pancoll separation solution (PanBiotech, Aidenbach, Germany). Magnetic isolation of T-cells was accomplished using the Pan T-cell Isolation Kit from Miltenyi Biotec (Bergisch Gladbach, Germany) according to the manufacturer’s instructions. After isolation, T-cells were incubated in RPMI complete medium supplemented with IL-2, further activated using T-Cell TransAct™ (Miltenyi Biotec, Bergisch Gladbach, Germany), and lentiviral transduced with the UniCAR construct as previously described [12]. The cells were subsequently cultured in TexMACS™ medium (Miltenyi Biotec, Bergisch Gladbach, Germany) supplemented with human IL-2, IL-7, and IL-15 (Miltenyi Biotec, Bergisch Gladbach, Germany). UniCAR T-cells were cultured in RPMI medium without cytokines for 24 h.

### 4.3. Peptide Synthesis—General Procedure

All peptides were synthesized by Fmoc-based solid-phase peptide synthesis (SPPS) using manual synthesis and a fully automated peptide synthesizer (Syro I, Multisyn-tech/Biotage, Charlotte, NC, USA). An amount of 50 mg of Rink Amide 4-methylbenzhydrylamine (MBHA) resin (100−200 mesh, loading: 0.78 mmol/g) was used as the solid support. Rink Amide MBHA resin was allowed to swell in 2 mL of dimethylformamide (DMF) for 15 min. Fluorenylmethyloxycarbonyl (Fmoc) group deprotection was achieved by treatment with 40% piperidine/DMF for 5 min, followed by treatment with 20% piperidine/DMF for 15 min. Fmoc-protected amino acids (5 eq.) were activated and coupled using 5 equiv of O-benzotriazole-*N*,*N*,*N*′,*N*′-tetramethyluronium-hexafluoro-phosphate (HBTU), 5 equiv of ethyl-2- cyano-2-(hydroxyimino) acetate (Oxyma), and 10 equiv of *N,N*′-diisopropylethylamine (DIPEA) over a 60 min time period followed by washing steps with DMF. Treatment with an acidic solution containing 90% TFA, 5% water, and 5% triisopropylsilane (TIPS) for 3 h 20 min induced cleavage of the assembled peptides from the resin with simultaneous deprotection of amino acid side chains. The resin was removed from the peptide solution through a syringe filter and ice-cold diethyl ether precipitated peptides were added. A syringe filter removed residual ether, and the precipitated crude peptides were dried under vacuum. Semipreparative HPLC purification, removing the solvent in a vacuum, and lyophilization delivered the pure peptides. Detailed protocols on the synthesis of E5B9-BBN2 monomeric and dimeric TMs as well as BBN2 dimer synthesis can be found in the Appendix A.

### 4.4. NOTA Modification of Ava-Glu-(Ava-BBN2)_2_

An amount of 3.5 mg (1 eq., 1.5 μmol) of Ava-Glu-(Ava-BBN2)_2_ and 0.9 mg (1.1 eq., 1.6 μmol) of *p*-SCN-Bn-NOTA was dissolved in 100 μL of DMF in a LoBind Eppendorf tube. The pH was adjusted to 9 by the addition of 3.1 μL (22.5 μmol) of 15 eq. triethylamine (TEA). The reaction mixture was incubated at 30 °C overnight before it was subjected to semipreparative HPLC purification (Method 2: t_R_ (NOTA-Ava-Glu-(Ava-BBN2)_2_ = 33.2 min). HPLC solvent was reduced under vacuum using a rotary evaporator, and lyophilization gave 3.4 mg (1.2 μmol, 81% isolated yield) of the chelator-modified peptide as a TFA salt (white powder). HPLC-QC (Method A-1): t_R_ (NOTA-Ava-Glu-(Ava-BBN2)_2_ = 18.5 min, 99% purity). MW C_134_H_204_N_34_O_29_S 2785.53, measured MALDI-MS (positive) *m*/*z* = 2788.0 [M+H]^+^, 2851.4 [M+Na+K]^+^.

### 4.5. Competitive Binding Assay

A competitive binding assay with ^125^I-Tyr^4^-BBN (PerkinElmer, Waltham, MA, USA) was used for the determination of the IC_50_ values (half-maximum concentration to inhibit binding of ^125^I-Tyr^4^-BBN(1-14) for NOTA-E5B9-BBN2 monomer or dimer. For this assay, PC-3 was maintained in 45% RPMI 1640, 45% Ham’s F-12, and 10% heat-inactivated fetal bovine serum (FBS). The cells were seeded in 6-well plates (5 × 10^5^ cells/well) and incubated overnight at 37 °C. The medium was then replaced with 1 mL of Dulbecco’s Modified Eagle Medium (DMEM) + 1% FBS containing increasing concentrations of NOTA-E5B9-BBN2 monomeric or dimeric TM in triplicates to reach a final concentration range from 5 pM to 0.5 μM. Then, ^125^I-Tyr^4^-BBN (0.05 nM final concentration) was added to each well, and the plates were incubated for 2 h at 4 °C. The cells were then rinsed twice with ice-cold phosphate-buffered saline (PBS) and harvested. The cells were placed on a Packard II gamma counter (PerkinElmer, Boston, MA, USA) to determine the cell-associated radioactivity. After decay correction, data were plotted using GraphPad Prism 9 Software (San Diego, CA, USA) using the sigmoidal dose–response equation, with counts per minute (cpm) of radioactivity bound versus the log of the concentration of NOTA-E5B9-BBN2 monomeric or dimeric TM for the determination of IC_50_ values. NOTA-E5B9-BBN2 dimeric TM was also tested in LNCaP cells. Gastrin-releasing peptide (GRP) served as the control.

### 4.6. Binding Assay Using Enzyme-Linked Immunosorbent Assay (ELISA)

Binding to the E5B9 and availability of this tag on the BBN2 TMs was assessed by ELISA using BD OptEIA™ Reagent Set B (BD Biosciences, Heidelberg, Germany) following the manufacturer’s instructions. High-binding 96-well plates (Corning, Berlin, Germany) were coated with different concentrations of BBN2 TMs, blocked with AD buffer, and incubated for 1 h at 37 °C with anti-E5B9 mAb. After washing, signal detection was performed using an anti-mouse IgG-HRP (Sigma-Aldrich, Darmstadt, Germany), and the optical density was measured at 450 nm using the Nanoquant Infinite M200 Pro (Tecan, Groedig, Austria).

### 4.7. Chromium-Release Cytotoxicity Assay

Standard chromium (^51^Cr)-release cytotoxicity assays were performed to assess the killing of tumor cells by redirected UniCAR T-cells as previously described [5]. Briefly, chromium-labeled target cells were incubated for 48 h with UniCAR T-cells at an effector-to-target cell (E:T) ratio of 5:1 in the absence or presence of different TM concentrations. The radioactive signal in the supernatants was measured using a MicroBeta Microplate Counter (PerkinElmer, Rodgau, Germany).

### 4.8. Cytokine-Release Assay

Cytokine concentration in cell-free supernatants was determined using the BD OptEIA™ Human IFN-γ, IL-2, TNF-α, and GM-CSF ELISA Sets (BD Biosciences, Heidelberg, Germany) according to the manufacturer’s instructions. The supernatants were harvested after 48 h of co-culturing UniCAR T-cells and target cells in the presence or absence of TMs.

### 4.9. T-Cell Activation, Exhaustion, and Memory Phenotype Using Flow Cytometry

To assess T-cell activation and exhaustion as well as memory phenotype, UniCAR T-cells were co-cultured with cancer cells at an E:T ratio of 5:1 in the presence or absence of TMs for 48 h and the corresponding expression of cell surface markers was assessed using flow cytometry. The activation and exhaustion panels included the following anti-human mAbs: CD69, CD279 (PD-1), CD366 (TIM-3), and CD223 (LAG-3) (Miltenyi Biotec GmbH, Bergisch Gladbach, Germany). The memory phenotyping was based on the anti-human mAbs CD62L (BioLegend GmbH) and CD45RO (Miltenyi Biotec GmbH, Bergisch Gladbach, Germany). After incubation with the respective mAbs, the stained cells were analyzed using the MACSQuant Analyzer 10 and MACSQuantify Software Version 3.0.2 (Miltenyi Biotec, Bergisch Gladbach, Germany).

### 4.10. ^68^Ga-Labeling of E5B9-NOTA-BBN2 Monomeric and Dimeric TMs, and NOTA-Ava-Glu- (Ava-BBN2)_2_ Dimer

Detailed methodology on the labeling of E5B9-NOTA-BBN2 monomeric and dimeric TMs as well as NOTA-Ava-Glu-(Ava-BBN2)_2_ dimer with the radionuclide ^68^Ga can be found in the Appendix A.

### 4.11. Metabolic Stability In Vivo

For metabolic stability studies in vivo, normal male BALB/c mice (body weight: 23−26 g, Charles River Laboratories, Saint-Constant, QC, Canada) were anesthetized through inhalation of isoflurane in 40% oxygen/60% nitrogen (gas flow 1 L/min) prior to i.v. radiotracer injection via the tail vein. Mice were injected with 28-50 MBq of E5B9-^68^Ga-NOTA-BBN2 TM, E5B9-^68^Ga-NOTA-(BBN2)_2_ TM, or ^68^Ga-NOTA-Ava-Glu-(BBN2)_2_ dimer in 150 μL of 10% EtOH/saline. Venous blood samples were collected at 5, 15, 30, and 60 min post-injection via the mouse tail vein and further processed. Blood cells were separated by centrifugation (13,000 rpm × 5 min). The supernatant was removed, and the contained proteins were precipitated by the addition of 2 volume parts of methanol (2 vol of MeOH/1 vol of the sample). Another centrifugation step (13,000 rpm × 5 min) was performed to obtain plasma. The clear plasma supernatant was injected into a Shimadzu HPLC system. The samples were analyzed using a Phenomenex Luna 10u C18(2) 100A, 250 × 4.6 mm column, at a constant flow rate of 1 mL/min, and the following gradient with water/0.2% TFA as solvent A and acetonitrile as solvent B: 0−3 min 10% B, 10 min 30% B, 17 min 50% B, 23 min 70% B, and 27−30 min 90% B. For each radiotracer (monomers and dimers with and without E5B9) blood samples from n = 3 mice were analyzed. A total of 3 mice were used multiple times for different radiotracers. Radiotracer experiments were recovery experiments with >1 µM compound concentrations injected.

### 4.12. Dynamic PET Imaging Studies

For PET experiments and as prostate cancer xenograft models, male PC3 and LNCaP tumor-bearing BALB/c Nude mice were used (body weight: 22−24 g, Charles River Laboratories, Saint-Constant, QC, Canada). To generate the tumor xenografts, about 5−6 × 10^6^ PC3 cells in 100 μL of PBS or 20−25 × 10^6^ LNCaP cells in 200 μL of PBS/Matrigel (50/50) were subcutaneously injected into the left shoulder of male BALB/c Nude mice. For the androgen-dependent LNCaP model, a 1.5 mg pellet containing dehydroepiandrosterone DHEA (60-day release; Innovative Research of America, Sarasota, FL, U.S.A.) was implanted subcutaneously into the upper right flank shortly before LNCaP cell injection. PC3 tumors and LNCaP tumors reached sizes of ∼300−500 mm^3^ after 3−5 weeks and 6−8 weeks, respectively, and were used for the in vivo experiments. PET imaging of E5B9-^68^Ga-NOTA-BBN2 monomeric and dimeric TMs was performed on an INVEON® PET/CT scanner (Siemens Preclinical Solutions, Knoxville, TN, USA). Prior to radiotracer injection, mice were anesthetized through inhalation of isoflurane in 40% oxygen/60% nitrogen (gas flow 1 L/min), and body temperature was kept constant at 37 °C. The mice were placed in a prone position in the center of the field of view. A transmission scan for attenuation correction was not acquired. Mice were injected with 4−8 MBq of E5B9-^68^Ga-NOTA-BBN2 or E5B9-^68^Ga-NOTA-(BBN2)_2_ in 150 μL 10% EtOH/saline solution through a tail vein catheter. For blocking studies, PC3- and LNCaP-tumor-bearing NSG mice were pre-dosed i.v. with 300 μg of Ava-BBN2 (15 min prior to radiotracer injection) in 50 uL saline. Data acquisition was performed over 60 min in 3D list mode. The dynamic list mode data were sorted into sinograms with 54 time frames (10 × 2, 8 × 5, 6 × 10, 6 × 20, 8 × 60, 10 × 120, 6 × 300 s). The frames were reconstructed using maximum a posteriori (MAP) as the reconstruction mode. No correction for partial volume effects was applied. The image files were processed using the ROVER v2.0.51 software (ABX GmbH, Radeberg, Germany). Masks defining 3D regions of interest (ROI) were set, and the ROIs were defined by thresholding. ROIs covered all visible tumor mass of the subcutaneous tumors, and the thresholds were defined by 50% of the maximum radioactivity uptake level. Mean standardized uptake values [SUVmean = (activity/mL tissue)/(injected activity/body weight), μL/g] were calculated for each ROI, and time−activity curves (TACs) were generated. For each experimental group (different radiotracers and mouse tumor models as well as control and blocking experiments), n = 3–6 mice were measured. A total of 9 PC3 tumor-bearing mice and a total of 7 LNCaP tumor-bearing mice were used multiple times for different radiotracers as well as control and blocking experiments. Radiotracer experiments are recovery experiments with >1 µM compound concentrations injected (except blocking experiments, which represent final experiments). Blocking experiments were conducted in the same mice: first, a control scan was measured and then repeated 1–2 days after in the presence of a blocking concentration of 300 μg of Ava-BBN2. All semi-quantified PET data are presented as means ± SEM. Statistical differences were tested by Student’s *t*-test and the results were considered significant for *p* < 0.05.

### 4.13. Statistical Evaluation

Statistical analysis was performed using GraphPad Prism 9.0 (GraphPad Software) and statistical significance was determined using either unpaired *t*-test, one-way ANOVA with Dunnett’s multiple comparison test, or two-way ANOVA with Turkey’s multiple comparison. *p* values below 0.05 were considered significant as follows: *p* < 0.05 (*), *p* < 0.01 (**), *p* < 0.001 (***), and *p* < 0.0001 (****).

## 5. Conclusions

The present study demonstrates that the E5B9-BBN2 TMs, in particular the dimeric BBN2 TM, are viable and promising immunotheranostic agents in preclinical prostate cancer models. In combination with UniCAR T-cells, these TMs have proven to efficiently eradicate such cancer cells, additionally presenting excellent in vivo metabolic stability and low immunogenicity. Such a combined diagnostic imaging and treatment approach potentiates future advances in personalized medicine for prostate cancer patients, especially for PSMA-negative metastatic disease.

## Figures and Tables

**Figure 1 ijms-26-02686-f001:**
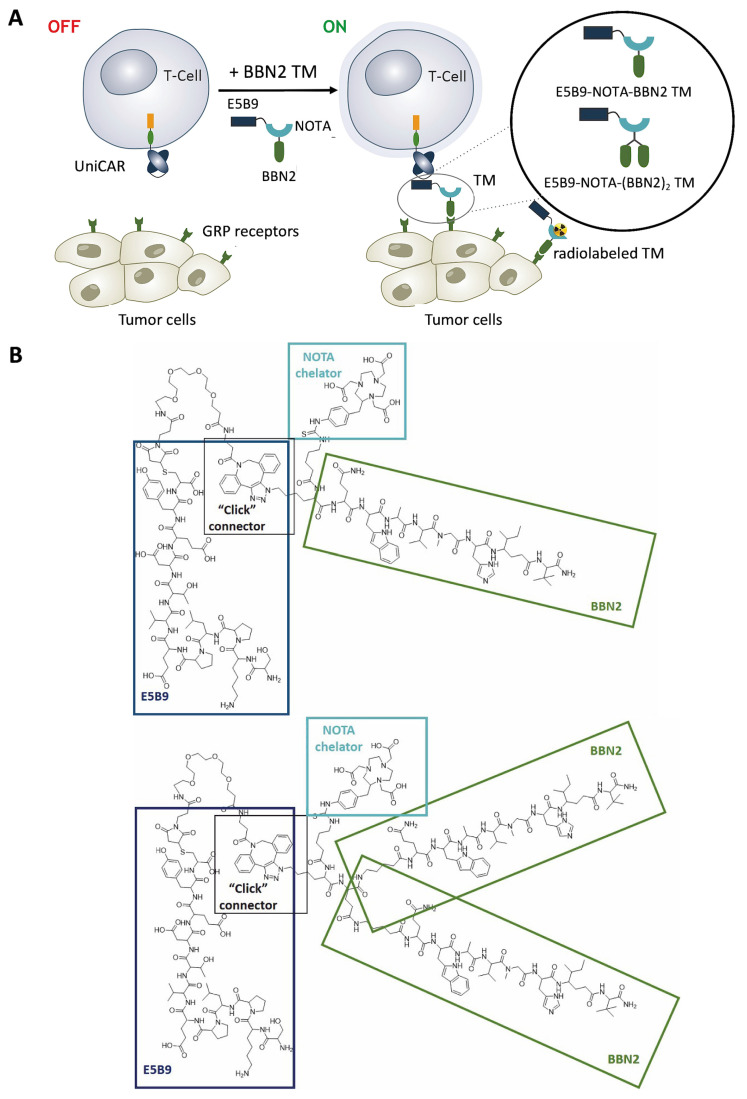
Schematic representation of the UniCAR system targeting the GRP receptor using the monomeric and dimeric BBN2 TMs (**A**); peptide structures of the synthesized E5B9-NOTA-BBN2 monomeric and dimeric TMs (**B**).

**Figure 2 ijms-26-02686-f002:**
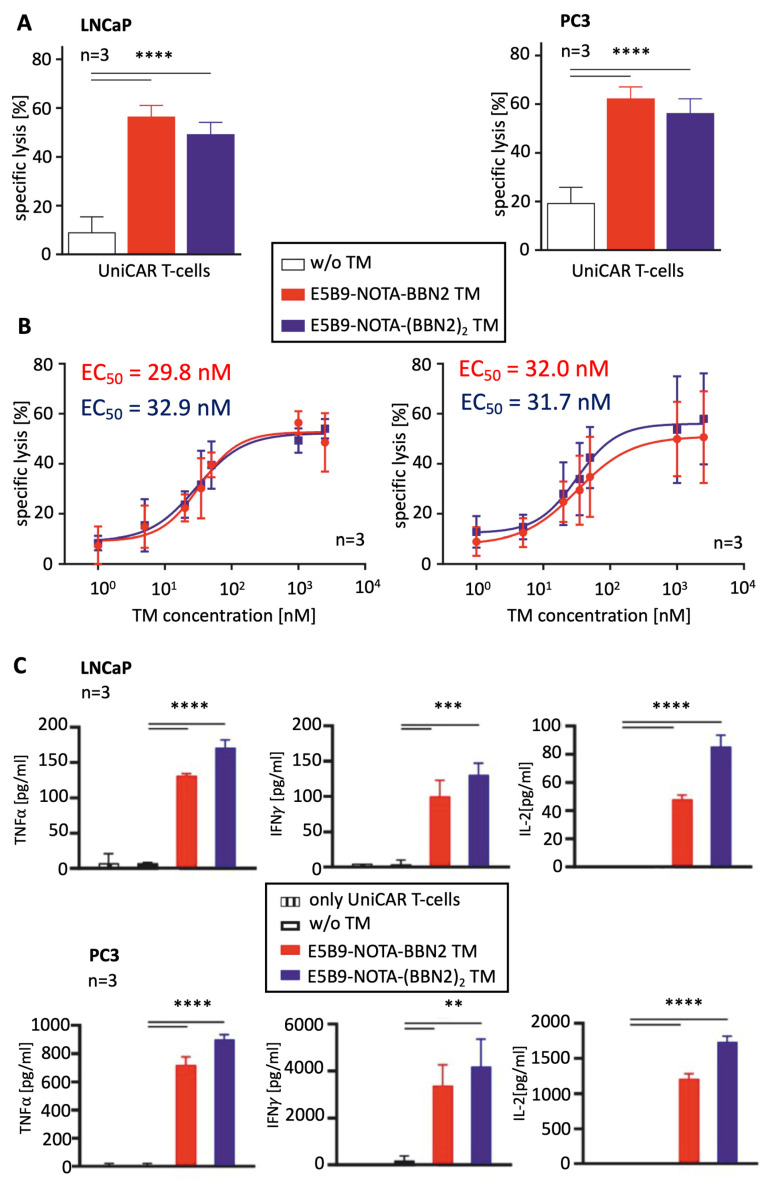
Cytotoxicity assessment (**A**), EC_50_ determination (**B**), and pro-inflammatory cytokine profile (**C**) of UniCAR T-cells redirected by monomeric and dimeric BBN2 TMs against PC3 and LNCaP cancer cells. Data are shown as mean ± SEM. EC_50_: half-maximum effective concentration. *p* < 0.01 (**), *p* < 0.001 (***), and *p* < 0.0001 (****).

**Figure 3 ijms-26-02686-f003:**
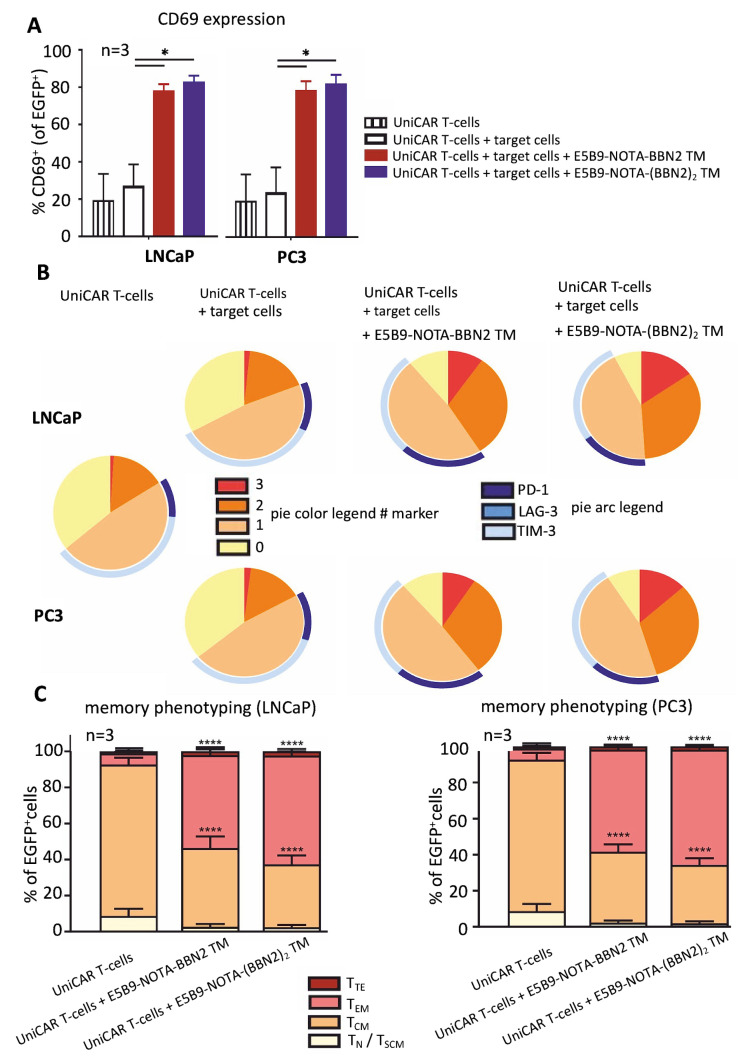
Activation (**A**), exhaustion (**B**), and memory (**C**) phenotyping of UniCAR T-cells redirected by BBN2 TMs to target PC3 or LNCaP cancer cells. Data are shown as mean ± SEM. *p* < 0.05 (*), *p* < 0.0001 (****).

**Figure 4 ijms-26-02686-f004:**
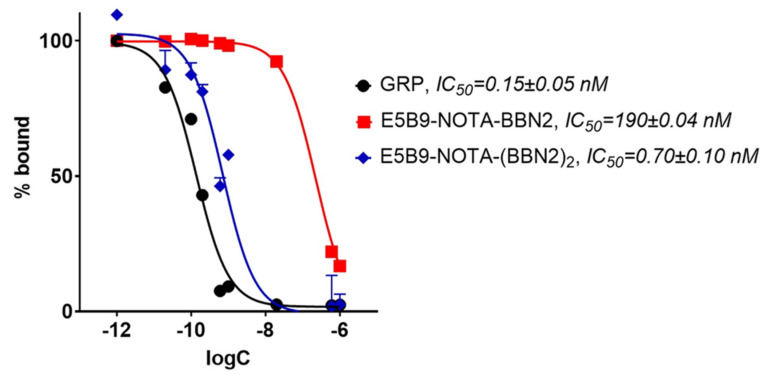
In vitro competitive cell binding of E5B9-NOTA-BBN2 monomeric and dimeric TMs in PC3 cells in comparison to reference gastrin-releasing peptide (GRP). Data are shown as mean ± SEM. IC_50_: half-maximum inhibition concentration.

**Figure 5 ijms-26-02686-f005:**
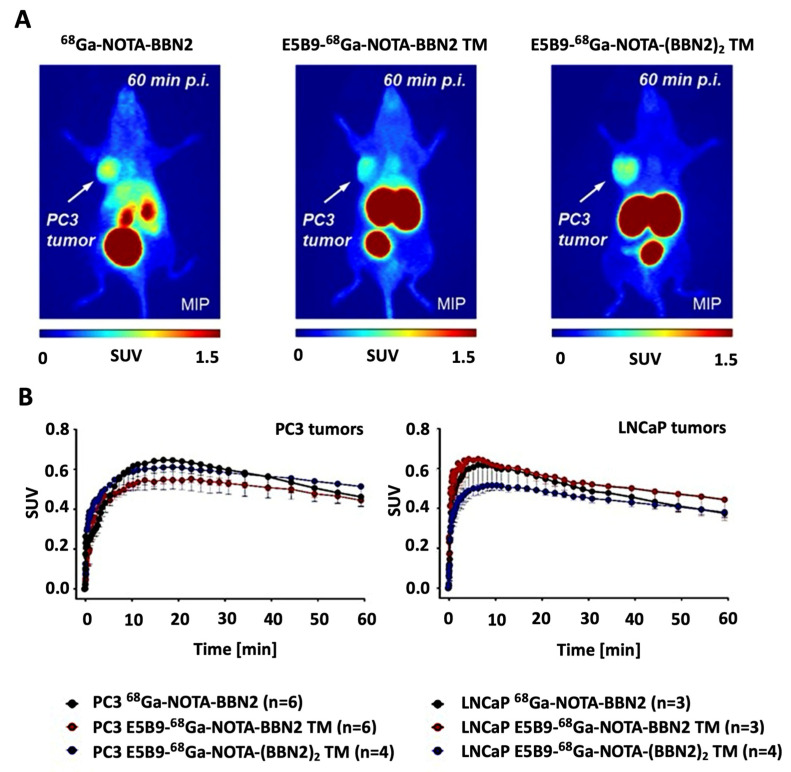
Representative PET images after injection of ^68^Ga-labeled BBN2 TMs and ^68^Ga-NOTA-BBN2 into PC3 tumor-bearing mice at 60 min p.i. (**A**). Time–activity curves (TACs) for tumor uptake profiles of ^68^Ga-labeled monomeric and dimeric TMs versus ^68^Ga-NOTA-BBN2 alone (**B**) in PC3 tumors and LNCaP tumors. Data are shown as SUV*_mean_* and mean ± SEM.

**Figure 6 ijms-26-02686-f006:**
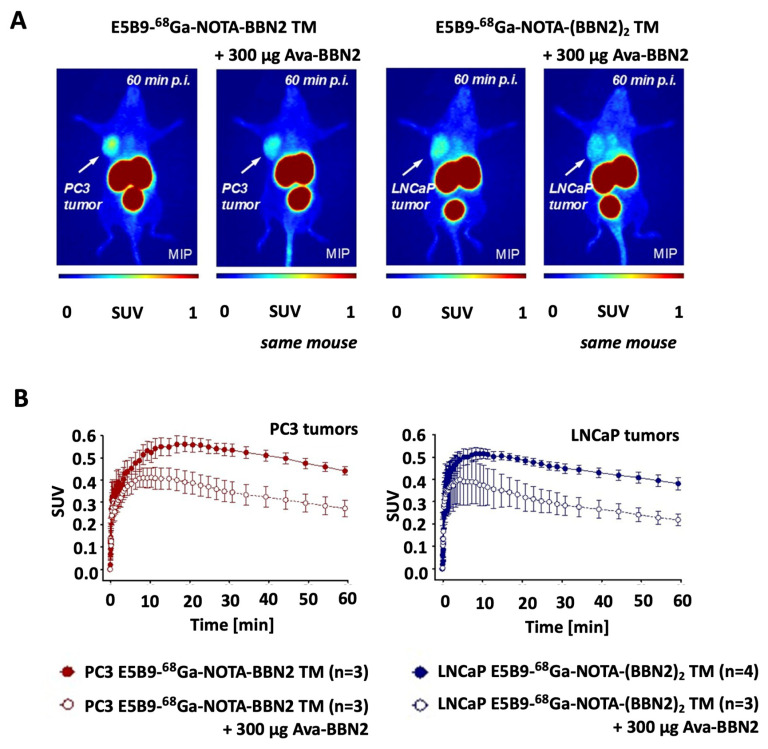
Representative PET images of in vivo blocking of ^68^Ga-NOTA-E5B9-BBN2 TM in a PC3 tumor-bearing mouse and ^68^Ga-labeled E5B9-(BBN2)_2_ TM in a LNCaP tumor-bearing mouse in the absence and presence of 300 µg of Ava-BBN2 (**A**) and their respective TACs for tumor uptake profiles (**B**). Data are shown as SUV*_mean_* and mean ± SEM.

**Figure 7 ijms-26-02686-f007:**
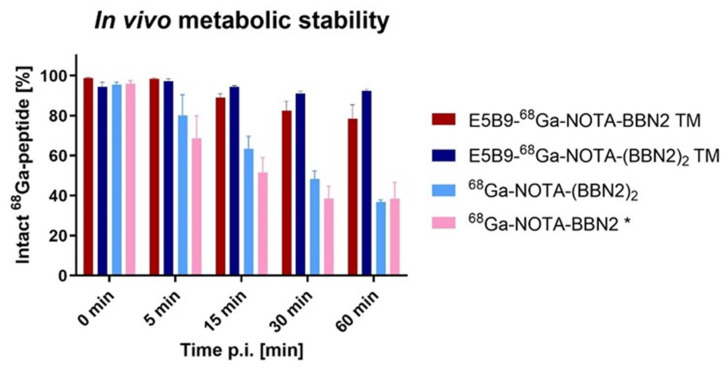
In vivo metabolic stability of ^68^Ga-labeled peptides E5B9-NOTA-BBN2 TM, E5B9-NOTA-(BBN2)_2_ TM, NOTA-(BBN2)_2_, and NOTA-BBN2 in venous blood plasma. Data are shown as % of the intact peptide after 5 to 60 min p.i. and as mean ± SEM (n = 3). * Data from Richter et al. [30].

## Data Availability

All data relevant to this study are included in the article or in the Appendix A.

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
