# Peer review of "Tackling Prostate Cancer with Theranostic E5B9-Bombesin Target Modules (TMs): From Imaging to Treatment with UniCAR T-Cells"

_ijms, 2025, doi:10.3390/ijms26062686_

Round 1

Reviewer 1 Report

Comments and Suggestions for Authors

In this paper, authors investigated the in vivo dynamics of functional molecules (target modules) that are expected to play an important role in UniCAR-T cell therapy by labeling them with PET nuclide Ga-68. This study has produced beneficial results that scientifically support the future potential of CAR-T cell therapy. There are no problems with the content of the paper, and reviewer considers it is worth of being published in IJMS. Just one request is that all figures and graphs are so small and not of high resolution, therefore, please correct them appropriately.

Author Response

Reviewer Comment:

In this paper, authors investigated the in vivo dynamics of functional molecules (target modules) that are expected to play an important role in UniCAR-T cell therapy by labeling them with PET nuclide Ga-68. This study has produced beneficial results that scientifically support the future potential of CAR-T cell therapy. There are no problems with the content of the paper, and reviewer considers it is worth of being published in IJMS. Just one request is that all figures and graphs are so small and not of high resolution, therefore, please correct them appropriately.

Response: We thank the reviewer for his careful review and valuable comments. We have now improved the size and quality of all figures for our manuscript.

Reviewer 2 Report

Comments and Suggestions for Authors

Dear Authors,

Congratulations on your work! The study presents the combined use of the UniCar peptide epitope E5B9 with the GRPR-targeting peptide bombesin antagonist BBN2 in monomeric and dimeric forms for imaging of prostate cancers. This study encompasses the synthesis processes, as well as all in vitro studies for the characterization of the compounds and in vivo evaluation using two different prostate cancer models.

The manuscript is written in a clear and precise manner, and I have only some considerations:

A) Major revision:

1) I understand that the aim of the article was to assess the specificity and therapeutic potential of the BBN2 TMs compounds through the induction of cell death in prostate cancer cell lines. However, it would be important to include, in addition to cells that express the GRPR receptor, a non-tumor cell line to evaluate the differences in the effects of the compounds, to be incorporated into the studies presented in Figure 2.

Minor revision:

1) For better comprehension, it is important that Figures 1, 2, and 3 be presented in a larger size.

2) In item 4.3, please correct treatment with an acidic solution containing 95% TFA, 5% water, 5% triisopropylsilane. The word triisopropylsilane is missing the letter r, and the percentages exceed 100%.

Author Response

Please find the point-by-point responses to the reviewer's comments below and attached as word file.

Reviewer 2 Comments:

Congratulations on your work! The study presents the combined use of the UniCAR peptide epitope E5B9 with the GRPR-targeting peptide bombesin antagonist BBN2 in monomeric and dimeric forms for imaging of prostate cancers. This study encompasses the synthesis processes, as well as all in vitro studies for the characterization of the compounds and in vivo evaluation using two different prostate cancer models.

The manuscript is written in a clear and precise manner, and I have only some considerations:

  1. A) Major revision:

1) I understand that the aim of the article was to assess the specificity and therapeutic potential of the BBN2 TMs compounds through the induction of cell death in prostate cancer cell lines. However, it would be important to include, in addition to cells that express the GRPR receptor, a non-tumor cell line to evaluate the differences in the effects of the compounds, to be incorporated into the studies presented in Figure 2.

Response: We thank the reviewer for his detailed review and valuable comment. In previous studies, we have demonstrated that antagonist bombesin peptide BBN2 targets GRPR specifically and with high sensitivity (Richter S, Wuest M, Bergman CN, Krieger S, Rogers BE, Wuest F. Mol Pharm. 2016;13(4):1347-57; Ferguson S, Wuest M, Richter S, Bergman C, Dufour J, Krys D, Simone J, Jans HS, Riauka T, Wuest F. Nucl Med Biol. 2020;90-91:74-83; Richter S, Wuest M, Bergman CN, Way JD, Krieger S, Rogers BE, Wuest F. Rerouting the metabolic pathway of (18)F-labeled peptides: the influence of prosthetic groups. Bioconjug Chem. 2015;26(2):201-12). The addition of peptide epitope E5B9 does not interfere with the BBN2 specificity for binding to GRPR as outlined in this study and also shown in previous work with different peptide or antibody-based E5B9 TMs (few selected examples: Loureiro, L.R.; Hoffmann, L.; Neuber, C. et al. Immunotheranostic target modules for imaging and navigation of UniCAR T-cells to strike FAP-expressing cells and the tumor microenvironment. J. Exp. Clin. Cancer Res. 2023, 42, 1–17; Albert, S.; Arndt, C.; Koristka, S. et al. From mono- to bivalent: improving theranostic properties of target modules for redirection of UniCAR T cells against EGFR-expressing tumor cells in vitro and in vivo. Oncotarget 2018, 9, 25597; Arndt, C.; Feldmann, A.; Koristka, S. et al. A theranostic PSMA ligand for PET imaging and retargeting of T cells expressing the universal chimeric antigen receptor UniCAR. Oncoimmunology 2019, 8, 1659095).

While GRPR is overexpressed in prostate cancers as well as others, its basic expression in normal tissue/non-cancerous cells is very low to almost negligible (Mansi R, Fleischmann A, Mäcke HR, Reubi JC. Targeting GRPR in urological cancers--from basic research to clinical application. Nat Rev Urol. 2013;10(4):235-44., Markwalder R, Reubi JC. Gastrin-releasing peptide receptors in the human prostate: relation to neoplastic transformation. Cancer Res. 1999;59(5):1152-9). While we have not yet used the TMs to target primary prostate cells, in previous studies we have performed related experiments targeting antigen-negative cells. In none of these experiments a non-specific targeting was observed, including in phase I clinical trials. To fulfill the request of the reviewer, we have tested the cytotoxicity of the novel TMs against the human embryonic kidney cell line HEK 293T, that is GRPR-negative. In line with our previous studies and as expected, the E5B9-BBN2 TMs do not kill these cells non-specifically. We prefer these negative results, not to include as an additional Figure. We have added the sentence in Section 2.2: ‘In line with previous control studies, the novel bombesin targeting TMs do not redirect UniCAR T cells to GRPR-negative cells (data not shown).’ To support this statement, however, we have included these data in the rebuttal report:

Minor revision:

1) For better comprehension, it is important that Figures 1, 2, and 3 be presented in a larger size.

Response: We thank the reviewer for the suggestion, and we have now improved the quality and size of Figures 1, 2, and 3 (and also Figures 5 and 6) in the manuscript.

2) In item 4.3, please correct treatment with an acidic solution containing 95% TFA, 5% water, 5% triisopropylsilane. The word triisopropylsilane is missing the letter r, and the percentages exceed 100%.

Response: We thank the reviewer for pointing that out. This was a typo on our end. We have corrected the items in the manuscript (‘…containing 90% TFA, 5% water, 5% triisopropylsilane’).

Reviewer 3 Report

Comments and Suggestions for Authors

In the current manuscript Loureiro et al. describe the synthesis and evaluation of two E5B9-bearing modules as means of PET imaging and immunotherapy of prostate cancer. Their constructs consist of the degradation resistant BB2 analogues, for recognition of GRPR on prostate cancer cells, the E5B9 motif for activation of T-cells and they are coupled with SCN-Bn-NOTA for labelling with Ga-68. The authors are presenting a thorough evaluation of their constructs with very promising and interesting results. The authors give detailed data and explanations behind the logic of their experiments and with their convincing data, a follow-up experimental therapy study would be most welcome.

Through the manuscript minor changes / addition should be made, which are given below (in no particular order):

  1. I would advise the authors to follow the guidelines for the presentation of radiopharmaceuticals and radionuclides in the text. It is essential for all the community to use the same way when presenting radiopharmaceuticals in order to avoid misunderstandings and misconceptions.
  2. Figures 1,2 and 3 need to be adjusted in size and with a higher resolution in order to be easier to read.
  3. In line 93 the authors write that they use NOTA as chelator, which is not correct. They used a derivative of NOTA, SCN-Bn-NOTA. In order to avoid any misunderstandings, I would advise the authors either to mention that they use SCN-Bn-NOTA which is going to be denoted as NOTA or to change the abbreviation.
  4. In line 183, there is a space missing, “p.i.in” should be “p.i. in”
  5. It would be a nice addition on the dataset if the authors could provide the IC50s for the metalated conjugates.
  6. In lines 190-191, the authors make note that the new analogues display a swift in their elimination pattern towards renal excretion. It would be nice to include these data from the dynamic PET imaging in their manuscript or in the supplementary. That will strengthen their statement and could be used for future reference.
  7. Also, the authors note that the addition of the E5B9 moiety in their analogues enhances in vivo stability, which they prove in the manuscript. Do they have any speculation why? Is it due to binding on cells / proteins in the blood stream or just the size of the E5B9 that sterically hinders the action of the enzymes? Blood kinetics for the dynamic PET scans could provide some insight on the matter.

Author Response

Please find the point-by-point responses to the reviewer's comments below and attached as word file.

Reviewer 3 Comments:

In the current manuscript Loureiro et al. describe the synthesis and evaluation of two E5B9-bearing modules as means of PET imaging and immunotherapy of prostate cancer. Their constructs consist of the degradation resistant BB2 analogues, for recognition of GRPR on prostate cancer cells, the E5B9 motif for activation of T-cells and they are coupled with SCN-Bn-NOTA for labelling with Ga-68. The authors are presenting a thorough evaluation of their constructs with very promising and interesting results. The authors give detailed data and explanations behind the logic of their experiments and with their convincing data, a follow-up experimental therapy study would be most welcome.

Through the manuscript minor changes / addition should be made, which are given below (in no particular order):

  1. I would advise the authors to follow the guidelines for the presentation of radiopharmaceuticals and radionuclides in the text. It is essential for all the community to use the same way when presenting radiopharmaceuticals in order to avoid misunderstandings and misconceptions.

Response: We carefully have checked the consensus nomenclature for radiopharmaceuticals by Coenen et al. 2017 and based on that have made changes in the introduction line 69: 'Gallium-68 (68Ga)-RM2', which is now stated as ‘68Ga-RM2’.

2. Figures 1,2 and 3 need to be adjusted in size and with a higher resolution in order to be easier to read.

Response: Figures 1, 2, and 3 (and in addition Figures 5 and 6) have now been improved in size and quality.

3. In line 93 the authors write that they use NOTA as chelator, which is not correct. They used a derivative of NOTA, SCN-Bn-NOTA. In order to avoid any misunderstandings, I would advise the authors either to mention that they use SCN-Bn-NOTA which is going to be denoted as NOTA or to change the abbreviation.

Response: We thank the reviewer for the comment. The manuscript now states ‘SCN-Bn-NOTA (denoted as NOTA)’ in line 93.

4. In line 183, there is a space missing, “p.i.in” should be “p.i. in”

Response: We have corrected this item accordingly.

5. It would be a nice addition on the dataset if the authors could provide the IC50s for the metalated conjugates.

Response: We thank the reviewer for his valuable comment. While the natGa-complexes of the E5B9-BBN2 TMs were not evaluated in the present study, we have demonstrated in the past that the metalated complexes natGa-DOTA-BBN2 (Ferguson S, Wuest M, Richter S, Bergman C, Dufour J, Krys D, Simone J, Jans HS, Riauka T, Wuest F. Nucl Med Biol. 2020;90-91:74-83) and natGa-NOTA-BBN2 (Richter S, Wuest M, Bergman CN, Krieger S, Rogers BE, Wuest F. Mol Pharm. 2016;13(4):1347-57) are very similar in binding to the GRPR as the non-metalated complexes: IC50(natGa-DOTA-BBN2) = 15±6 nM vs. IC50(DOTA-BBN2) = 43±5 nM and IC50(natGa-NOTA-BBN2) = 4.6±1.2 nM vs. IC50(NOTA-BBN2) = 8.2±0.2 nM. A similar trend in further improved binding would be expected for the metalated natGa-NOTA-BBN2-E5B9 TMs.

6. In lines 190-191, the authors make note that the new analogues display a swift in their elimination pattern towards renal excretion. It would be nice to include these data from the dynamic PET imaging in their manuscript or in the supplementary. That will strengthen their statement and could be used for future reference.

Response: We agree with the reviewer. We have included a new Figure into the Supplementary Data that demonstrates the kidney clearance parameters obtained from the dynamic PET data of 68Ga-NOTA-BBN2 vs. 68Ga-NOTA-BBN2-E5B9 displaying the shift towards more renal elimination when the E5B9 peptide epitope is attached to the BBN2 peptide structure:

7.Also, the authors note that the addition of the E5B9 moiety in their analogues enhances in vivo stability, which they prove in the manuscript. Do they have any speculation why? Is it due to binding on cells / proteins in the blood stream or just the size of the E5B9 that sterically hinders the action of the enzymes? Blood kinetics for the dynamic PET scans could provide some insight on the matter.

Response: We thank the reviewer for his valuable comment. Right now, we can only speculate how the peptide epitope E5B9 may improve resistance to extracellular proteases due to its amino acid sequence. It has been described that short protein motifs are able to stabilize peptides and small proteins. These motifs were rich in lysine residues (Rethi-Nagy Z, Abraham E, Udvardy K, Klement E, Darula Z, Pal M, Katona RL, Tubak V, Pali T, Kota Z, Sinka R, Udvardy A, Lipinszki Z. STABILON, a Novel Sequence Motif That Enhances the Expression and Accumulation of Intracellular and Secreted Proteins. Int J Mol Sci. 2022;23(15):8168) or proline residues (Walker JR, Altman RK, Warren JW, Altman E. Using protein-based motifs to stabilize peptides. J Pept Res. 2003;62(5):214-26). In line with these publications, the E5B9 peptide epitope carries 2 proline and 1 lysine residue. Those amino acids in E5B9 maybe key elements for improved in vivo stability. This has been added to the revised manuscript. However, future detailed studies will be necessary to shed light into this interesting observation.
